# Hydraulic Measuring Hoses as Pressure Signal Distortion—Mathematical Model and Results of Experimental Tests

**DOI:** 10.3390/s23167056

**Published:** 2023-08-09

**Authors:** Klaudiusz Klarecki, Dominik Rabsztyn

**Affiliations:** Department of Engineering Processes Automation and Integrated Manufacturing Systems, Faculty of Mechanical Engineering, Silesian University of Technology, 44-100 Gliwice, Poland

**Keywords:** signal distortion, pressure peak, hydraulic line dynamics, hardware filtration of pressure signals, hydrostatic drives

## Abstract

The article presents the results of a developed model and experimental studies of the Minimess^®^ hydraulic signal hose’s influence on the changes in the indications of the pressure transducer during the high dynamics of hydrostatic drives and controls. The model test results show that measuring hoses can be used as hardware low-pass filters during the digital recording of pressure waveforms. However, the cut-off frequency values of the measuring hoses obtained using the model are dramatically lower than those observed during the experiment. The experiment results show that the measuring hoses can only be used without any limitations to measure the average pressure value. In the case of measuring pressure waveforms, the user should carefully choose the measuring hose length. For this reason, the relationship between the measuring hose length and its cut-off frequency should be known.

## 1. Introduction

The further dynamic development of mechatronic systems is not possible without suitable effectors as these systems correctly perform mechanical interactions with the environment. As effectors are characterised by a favourable ratio of the developed force (or torque) to their weight or overall dimensions [1], there are solutions based on hydrostatic drives [2]. This means that the need to select highly effective force effectors in selected applications of mechatronic systems has created a new mechatronics subfield called hydrotronics. Hydrotronic hydrostatic drives require precise motion control and force control of hydrostatic actuators, which is a result of their integration with extensive microprocessor control systems, the incessant improvement of the static [3] and dynamic parameters [4] of basic components of actuator assemblies, such as proportional valves and servo valves [5], and sensors for measuring controlled hydraulic and mechanical quantities [6]. Hydrotronics is perceived as the present and future of fluid drive development in mechanical engineering. Hydrotronic drives, equipped with complex automatic control systems [7] and containing an advanced electro-hydraulic valve system [8], characterised by high dynamics of operation [6,9], very often operate in permanent transient states due to the user requirements. Such systems are characterised by the capability of very precise settings of position, speed, torque, or force, but pressure sensors are often used as the feedback elements in force (or torque) control loops, which is the easiest way to install them in a drive system.

In many cases, pressure sensors that are the main part of the automatic control system (e.g., the pressure force) are not directly installed on the supply connection of the receiver chamber (or chambers); this may be due to various reasons, e.g., facilitating service access to the sensors. In this case, pressure sensors are connected to selected points of the hydraulic system, such as cylinders or hydraulic motor connections, in which measuring test points are installed by using appropriate lengths of measuring hoses.

Measuring test points and measuring hoses (e.g., Minimess^®^ Test Points and Minimess^®^ hoses) [10] are handy tools to measure the operating pressures of hydraulic power packs or hydraulic drives. Because of this, they are very popular. In addition, measuring test points [11] can facilitate the easy deaeration of hydraulic elements and lines [12]. During the measurement, the end of the measuring hose is blinded by a pressure sensor. Due to the fact that the liquid does not flow through the hose, there is no average pressure loss, and its measured value is accurate.

The main reasons for the conducted research were encounters, as part of the authors’ industrial experience, with problems related to the precise control of the dynamic quantities (push or pull force, pressure difference) of the in servo drives of modern CNC hydraulic presses and full control of the kinematic and dynamic parameters in the main press drive. A thorough analysis of the design solutions adopted by the manufacturers of these presses showed that the pressure transducers were connected to the chambers of the main cylinder with hydraulic measuring hoses. The authors asked themselves: “How can measuring hoses affect the measurement of pressure in hydraulic systems?” Consequently, the following questions were formulated:Can this effect be neglected?Can users of hydraulic systems take advantage of this effect?

One possible answer to these questions, formulated by the authors, is the possibility of using measuring hoses as hardware low-pass filters during the digital acquisition of pressure courses. However, this way of using measuring hoses requires further research to determine the close relationship between the dimensions (especially length) of measuring hoses [13,14] and their cut-off frequency. This is particularly important for users of high-dynamic hydraulic systems which consist of servo valves or proportional control valves [8].

The article presents the developed mathematical model of damping hydraulic test leads and the results of experimental studies regarding different lengths of hydraulic Minimess^®^ measuring hoses [10].

## 2. Dynamic Phenomena Occurring in Hydraulic Lines

The processes related to the flow of the working medium through the conduits (flexible and rigid) in high-pressure hydraulic systems, obtained as a result of the forced or accidental operation of the drive system components, are related to the operational properties of the hydraulic system’s components, including maintaining the compressible liquid in the conduits, as dynamic hydraulic lines. The characteristics of non-stationary processes depend on the selected physical properties of the working liquid (for isothermal flow: density, viscosity, and compressibility β_0_) and the elasticity of the conduit and its geometrical dimensions (diameter and length) [14,15,16,17].

The values that were mentioned consist of the following hydraulic line parameters:Resistance, *R*_0_ (active resistance), taking into account the friction between liquid particles;Inertia, M_0_ (passive resistance), taking into account the inertia of the liquid;Capacitance, *C*_0_, which consists of the liquid compressibility, β_0_, and the elasticity of the wire line material.

The combined effects of resistance, inertia, and capacitance in dynamic conditions affect the distortions and delays of the waveforms occurring in the flow conduits of hydraulic drive systems and change their dynamic properties. The dynamic properties of wires are created by two possible methods:Using distributed parameters;Using focused parameters.

The dynamic properties of the line, determined by using distributed parameters, are created by using the combined, in a series of unlimited elements of the units in a single series, impedance, Z_0_, (considering resistance, *R*_0_, and inertia, M_0_) and the shunt admittance, Y_0_, (considering capacitance, *C*_0_).

In the presented method, the movement of the liquid is described using hyperbolic partial equations involving the wave phenomena based on the flow of the liquid in the wire [14,15,16,17].

The method of modelling the dynamic properties of the line by using the focused parameters assumes that the constant parameters of the wire (*R*_0_, M_0_, *C*_0_) are concentrated in appropriate places along the axis of the wire’s line. In other words, the description of the mathematical dynamic phenomena is presented using ordinary differential equations. These models are described in detail in [13,14,15,16,17,18,19,20,21,22,23,24,25].

## 3. Mathematical Model

During the analysis of the phenomena related to dynamic changes in pressure, special attention should be paid to the phenomena accompanying the hydraulic long line (HLL). A hydraulic line should be considered long when the length of the line is equal to or greater than the length of the pressure wave propagated in it. In [15], it was recommended to treat a hydraulic line as long if it meets the following condition:(1)$$\lambda_{f}\leq \frac{c_{0}}{10\cdot f_{max}}$$
where:*λ_f_*—Length of the hydraulic line {m};*c*_0_—Velocity of the pressure wave propagation (phase velocity) in the hydraulic conduit {m/s};*f_max_*—Maximum excitation frequency {Hz}.

If the condition is met, then the line is treated as an element with distributed parameters. Then, the changes in the pressure, p, and flow rate, Q, that propagate along the duct axis with a certain speed in the form of travelling and reflected waves should be taken into account [14,15]. In this case, one should consider the possibility of resonance phenomena resulting in an increase in the intensity of pressure pulsation amplitudes.

In the literature devoted to hydraulic long transmission line modelling in hydraulic systems [14,15,16,17], the two most commonly used methods of describing transient or quasi-steady waveforms have been described:Frequency method;Method of studying transient processes as a function of time.

The model studies conducted as part of the article are focused on the analysis of the dynamic phenomena of the hydraulic line using the frequency method [15].

In effect, the measuring hose is a small diameter pipe with high hydraulic resistance for the hydraulic fluid and a hydraulic long transmission line for the pressure signals. In accordance with [14,15,16,17], the hydraulic long transmission line can be described as a two-port network with two pairs of terminals connected to external circuits. A typical hydraulic two-port network is graphically presented in Figure 1.

In hydraulic long transmission lines, it is traditionally assumed that the distortion (signal) derived from the receiver is transferred to the hydraulic power pack. Obviously, the direction of fluid flow is reversed and occurs from the pump to the hydraulic receiver.

The relationship between the input and output ports is described by the matrix equation:(2)$$\left[\begin{array}{c} p_{\mathrm{in}}\\ q_{\mathrm{in}} \end{array}\right]=H\left(j\omega \right)\left[\begin{array}{c} p_{\mathrm{out}}\\ q_{\mathrm{out}} \end{array}\right]$$
where:$$H\left(j\omega \right)=\left[\begin{array}{cc} h_{11} & h_{12}\\ h_{21} & h_{22} \end{array}\right]$$
In accordance with [15,16,17], elements of the transfer function *H*(*jω*) can be described as:$$h_{11}=\cosh\left({T\psi }_{Z}j\omega \right)$$
$$h_{12}=Z_{c}\psi_{Z}\sinh\left({T\psi }_{Z}j\omega \right)$$
$$h_{21}=\frac{1}{Z_{c}\psi_{Z}}\sinh\left({T\psi }_{Z}j\omega \right)$$
$$h_{22}=\cosh\left({T\psi }_{Z}j\omega \right)$$
thereby:$$T=\frac{L}{c_{0}}$$
$$Z_{c}=\frac{\rho_{0}c_{0}}{{\pi R}^{2}}$$
where:*ψ_Z_*—Viscosity function;*T*—Time constant;*Z_c_*—The characteristic impedance of the pipe (hose) with radius, *R,* and length, *L*;*c*_0_—The speed of sound in the liquid inside the pipe (hose);*ρ*_0_—Fluid density.

In the measuring hose, the model of an adequate two-port network can be assumed as the object presented in Figure 2. The pressure transducer is mounted on the output port, so the output volumetric flow rate, *q*_out_, is neglected.

For the measuring hoses, the directions of fluid flow and signal propagation are consistent. In this case, the relationship between the input port and output port is described as:(3)$$\left[\begin{array}{c} p_{\mathrm{out}}\\ q_{\mathrm{out}} \end{array}\right]=H^{-1}\left(j\omega \right)\left[\begin{array}{c} p_{\mathrm{in}}\\ q_{\mathrm{in}} \end{array}\right]$$

In accordance with Equation (4), the output pressure, *p*_out_, can be described as:(4)$$p_{out}=\frac{1}{h_{11}\cdot h_{22}-h_{12}\cdot h_{21}}\left(h_{21}\cdot p_{in}-h_{12}\cdot q_{in}\right)$$

Equation (5) is useless due to the impossibility of estimating the inlet flow rate, *q*_in_.

The assumption of the high stiffness of the measuring element (diaphragm or substitute element) of the pressure transducer and the small volume of the measuring chamber allows for the assumption of a negligible flow rate at the connection of the measuring hose with the pressure transducer, *q*_out_ = 0.

For *q*_out_ = 0, there is no need to set an inverse transfer function. The problem of calculating the transfer function between the pressures *p*_in_ and *p*_out_ is reduced to the formula:(5)$$p_{in}=h_{11}\cdot p_{out}+h_{12}\cdot q_{out}=h_{11}\cdot p_{out}$$

Therefore, the pressure transfer function is:(6)$$\begin{array}{c} G_{p}\left(j\omega \right)=\frac{p_{\mathrm{out}}}{p_{\mathrm{in}}}=\frac{1}{h_{11}}\\ G_{p}\left(j\omega \right)=\frac{1}{\cos h\left({T\psi }_{Z}j\omega \right)} \end{array}$$

For laminar flow, the viscosity function can be accepted as:(7)$$\psi_{Z}=\frac{\frac{\sqrt{2}}{2}\Omega \left(\sqrt{-1+\sqrt{1+{\left(\frac{R_{0}}{\Omega }\right)}^{2}}}+j\sqrt{1+\sqrt{1+{\left(\frac{R_{0}}{\Omega }\right)}^{2}}}\right)}{j\Omega }$$
wherein Ω is a dimensionless frequency:(8)$$\Omega =\frac{{\omega R}^{2}}{\nu }$$
where:*ω*—Pressure signal frequency {rad/s};*ν*—Kinematic viscosity of the hydraulic fluid {m^2^/s};*R*_0_—Hydraulic resistance according to the Hagen–Poiseuille equation (laminar flow);*R*—Inner radius of the measuring hose.

The frequency response characteristic curves for the accepted hoses were calculated from Equations (6) and (7) using FreeMat software. Numerical calculations were performed for three lengths of the measuring hose, namely, DN2 (internal diameter 2 mm): 0.4 m; 0.8 m; and 1.5 m, and for four pressure values: 50 bar; 100 bar; 150 bar; and 200 bar. The internal radius of the hose is 1 mm, and the kinematic viscosity was accepted as the viscosity of HLP46 fluid at 30°. Additionally, calculations were performed for DN4 hoses (internal diameter 4 mm).

The results obtained for the DN2 measuring hoses are shown in Figure 3 and for DN4 hoses in Figure 4.

The results of calculations indicate that the pressure measurement using sensors installed at the end of the measuring hose can demonstrate very strong distortion. The cause of distortion is the fact that the measuring hose acts as a low-pass filter. The pressure signals at low frequencies are transmitted without distortion. However, the pressure signals above the cut-off frequency of the measuring hose are strongly damped. It is interesting that the damping per decade is not constant and increases for higher frequencies. The internal diameter of the measuring hoses is the most important factor for their cut-off frequencies. Comparing the results from Figure 3 and Figure 4, one can notice that a diameter twice as large is the cause of the enormous shift of the cut-off frequency, which was shifted about two decades. The cut-off frequency of the measuring hose was accepted as the pressure amplitude −3 dB, with this value corresponding approximately to an amplitude ratio equal to 0.7.

In addition, the cut-off frequency of the measuring hoses strongly depends on their lengths and only slightly depends on the average pressure. The properties of the measurement hoses are dependent on the average pressure because the pressure affects the propagation speed of sound in hydraulic hoses.

From the obtained results, it can be assumed that the measuring hoses can be used as the hardware low-pass filters during the digital acquisition of the pressure waveforms. The slopes of the frequency performance curves of the measuring hoses are not constant, as in typical low-pass filters. The slope for the first range of each curve of the pressure transfer function equals −20 dB/decade, so these values are nearly first-order low-pass filters.

The cut-off frequencies of DN2 hoses were as follows:~0.2 Hz for the 0.4 m long measuring hose;~0.05 Hz for the 0.8 m long measuring hose;Below 0.01 Hz for the 1.5 m long measuring hose.

The cut-off frequencies of DN4 hoses were as follows:


~15 Hz for the 0.4 m long measuring hose;~4 Hz for the 0.8 m long measuring hose;~1 Hz for the 1.5 m long measuring hose.


The calculation results for models of the measuring hoses based on the mathematical model of a hydraulic long transmission line are controversial and therefore require experimental verification.

## 4. Assumptions of the Experimental Test

A fundamental issue in the experimental research of the transfer function is the determination of the type of test signal to be provided for the test object. The most convenient forms of test signals are the Dirac delta function and white noise. For these kinds of test signals, the following should be recorded: the pressure waveforms at the input and output hose. The next step is the calculation of the transfer function using the FFT method. Another type is the harmonic signal. For harmonic signals, the analysis of the transfer function can be figured directly using the results in the time domain. The disadvantage of this method is the large amount of work resulting from the need to repeat the test many times for successive frequency settings. The above-mentioned types of test signals require the construction of a special hydraulic signal generator.

## 5. Periodic Changes in Pressure on the Discharge Line as Excitation Signals

What is the most convenient signal generator for the workshop testing of the pressure transfer function? The simplest test signal generator may be a positive displacement pump with significant flow pulsation. Therefore, it is best to use an external gear pump powered by a variable rotational speed motor. Additionally, the pressure value on the discharge line can be realised as a rectangular course by switching the flow between two lines with different hydraulic resistance values. On one of these lines, a pressure relief valve is installed; thus, in the switching moment, additional flow pressure peaks and oscillations of pressure occur. Two different values of pressure were obtained by switching the flow between two valves: the pressure relief valve and the throttle valve. The settings of valves were adopted so that the pressure changed by approx. 20 bar. The hydraulic test stand for preliminary research on the pressure transfer function of measuring hoses is presented in Figure 5.

The main elements of the test stand include an external gear pump type PGP511 0080, pressure relief valve type ASO4G2 (set to 215 bar), pressure line with measuring test point type EMA3/1/4ED, throttle valve type 9N600S, pressure relief valve type DBD-6, measuring hose DN2 type SMA3, two pressure transducers type HDA 4748-H-0250, and a portable data recorder type HMG 3010 [7,26].

It was assumed that research was to be carried out for:Three lengths of measuring hoses: 0.4 m, 0.8 m, and 1.5 m;Three values of average pressure: 50 bar, 100 bar, and 150 bar.

The recorded digital pressure signals were used as input data for the FFT method. The sampling frequency for the recording signals was accepted as 10 kHz. The cut-off frequency of pressure transducers was lower than 2 kHz; therefore, the pressure sensors acted as low-pass filters to prevent the aliasing phenomenon. The pressure transfer functions of the measuring hoses were determined using the FFT method (with a flat-top window width of 8192 points). The calculation results of the pressure transfer function using FFT were noisy despite the averaging of data from five FFT windows. For FFT results, the values of the measuring hose gain were calculated according to the relation:(9)$$G_{\mathrm{hose}}\left(f\right)=20\log\left|H\left(j\omega /2\pi \right)\right|$$
where:*G*_hose_(*f*)—The gain of the measuring hose (amplification of pressure pulsation by the measuring hose) {dB};*H*—The transfer function of the measuring hose in the frequency domain.

The selected results of this research step are presented in Figure 6. The courses of the pressure transfer functions in the frequency domain, which are shown in these figures, are almost smooth in the range from 0 Hz to 200 Hz. Above a frequency of 200 Hz, one can notice significant abrupt changes in the value of the transfer function of measuring hoses, which undermines the credibility of the results. Due to this factor, the analysis of these results is limited to the frequency range from 0 to 200 Hz.

The measuring hoses 0.4 m and 0.8 m in length transmit pressure signals to the frequency of approx. 100 Hz without distortion. Above 100 Hz, the hose of 0.4 m length acts as an amplifier of pressure pulsation, while the hose of 0.8 m length acts as a low-pass filter. The hose with a length of 1.5 m also acts as a low-pass filter for pressure signals, i.e., it can transmit undistorted pressure signals with a frequency below 10 Hz. Its cut-off frequency is approx. 65 Hz, and the slope is about −6 dB/decade (for 10 Hz–100 Hz range).

The phenomenon of amplification pressure pulsation in the short measuring hoses requires further explanation.

The first issue is whether the excitation signal was properly chosen. In Figure 7, one can see a spectrum of the recorded pressure in a measuring point mounted directly on the discharge line. A high level of the excitation signal is observed in the range from 1 to 30 Hz. In the frequency range from 30 to approx. 100 Hz, the excitation signal is a mediocre value, and above 100 Hz, the signal value approaches a value of the quantisation error of the measuring and data-logging device type HMG3010. The quantisation error of the measuring system consisting of a device type HMG3010 and a pressure sensor type HDA 4748-H-0250 was approx. 0.06 bar.

Additionally, a strong excitation signal at a frequency of 300 Hz was noted. This is due to the periodic nature of the gear pump action. Considering the above comments, it is clear that the adopted method of the excitation signal implementation was not appropriate to examine the pressure transfer function of the measuring hoses.

The recorded pressure chatters for high frequencies are due to the fact that the amplitudes of the high-frequency components were very low, with values close to the error of the applied pressure measurement system (Figure 7). The described phenomenon was the result of the adopted method of excitation of the tested object, which, as it turned out, was not suitable for testing the response of the test lead in the high frequency range. For this reason, in a further part of the research, an original method of object excitation in a wide frequency range was adopted, adapted from modal analysis methods (Impact Hammer Test).

## 6. Impact Hammer Test for Measuring Hoses

Impact hammer testing is a very popular and convenient method for modal tests of mechanical structures. The excitation signal resulting from the impact hammer is the physical realisation of the perfect impulse signal.

The hydraulic test stand for the impact research of the pressure transfer function of measuring hoses is presented in Figure 8.

The main elements of the test stand include a hydraulic power unit, as presented in Figure 5, a hydraulic cylinder Ø40/28 × 500 type HMI, ball valves type BVHP for tight closing of the working chamber of the cylinder, a hammer with rubber tips, a measuring hose DN2 type SMA3, two pressure transducers type HDA 4748-H-0250, and a portable data recorder type HMG3010 [7].

The hardness of the impact hammer tip determines the impulse shape (amplitude and duration) and the excitation bandwidth. The research was carried out for two rubber tips with different hardness. The tip type of the impulse hammer influences the form of the excitation signal; as shown in Figure 5. The sampling frequency of the pressure signal in the measuring point was 10 kHz and 100,000 samples were saved. The excitation signal was analysed using the FFT method with a flat-top window width of 16,384 samples. The results were averaged from three FFT windows. When a soft hammer tip was used, an almost uniform spectral distribution of the excitation signal was obtained in the frequency range from 1 to 100 Hz. This is very convenient for modal tests and the derivatives thereof. The only drawback of the excitation signals generated by the soft hammer tip is the low value of the excitation signal. For this reason, the research results for both hammer tips will be presented further in the article; Figure 9.

The spectrum of excitation signal during the Impact Hammer Test (namely, the pressure recorded on the input of the measuring hose) for the average pressure of 100 bar for both hardnesses of the hammer tip is shown in Figure 10.

For both test series, similar values of cut-off frequencies (−3 dB) were obtained, namely, approx.:160 Hz for the measuring hose 0.4 m in length;140 Hz for the measuring hose 0.8 m in length;50 Hz for the measuring hose 1.5 m in length.

The results of the tests considering the cut-off frequency of the measuring hoses using both methods are similar. The essential difference is that during the tests using the impact impulses as excitation signals, the amplification of the pressure pulsation was not observed. Each measuring hose acted like a low-pass filter with a similar slope, approx. −12 dB/decade. Therefore, the assumption that the pressure pulsations were strongly damped by measuring hoses is not completely true. The authors hoped that the dumping of the measuring hoses would be at least −20 dB per decade, according to Figure 3a.

## 7. Conclusions

The results of the numerical simulations based on the model of a hydraulic long transmission line partially confirmed the intuitive expectations that measuring hoses perform the function of low-pass filters for pressure signals. The extremely low values of the cut-off frequency for hoses with the nominal size DN2 were a great surprise. One of the potential causes of such results may be the unsuitable conditioning of the mathematical model of the hydraulic line of a model where the flow occurred at the end of the line and was adapted to a case of zero flow rate at the end. Without experimental verification of the obtained results, it is impossible to receive a clear answer regarding what might be the cause of the results obtained from the numerical calculations.

The experimental test results partially confirmed that it is possible to use the measuring hoses as hardware low-pass filters during the digital recording of pressure waveforms. The attenuation of the measuring hoses is not high and, for this reason, the selection of the hoses for conditioning the pressure signal during recording has to consider that hose cut-off frequencies should be significantly lower than half of the signal sampling frequency. It is noted further that the values of the cut-off frequency of the measurement hoses non-linearly depend on their lengths.

The research described in the article can indeed be treated as more utilitarian than basic. However, it can also be treated as a contribution to the discussion on the applicability of the mathematical model of the hydraulic long transmission line which is adopted in many literature positions.

## Figures and Tables

**Figure 1 sensors-23-07056-f001:**
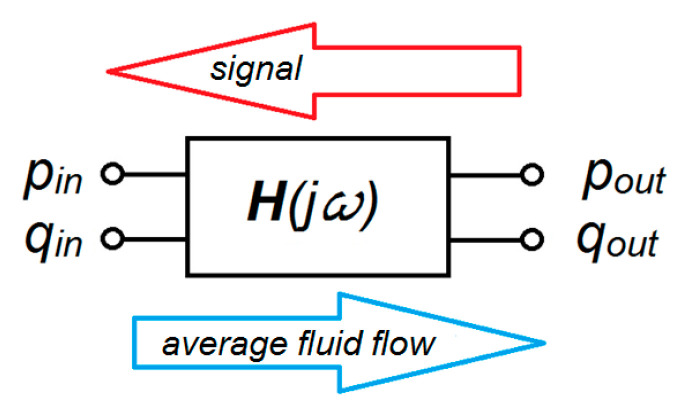
Typical hydraulic two-port network.

**Figure 2 sensors-23-07056-f002:**
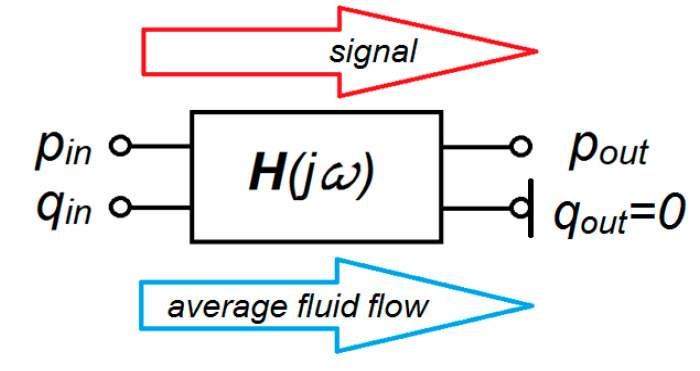
Measuring hose as a two-port element.

**Figure 3 sensors-23-07056-f003:**
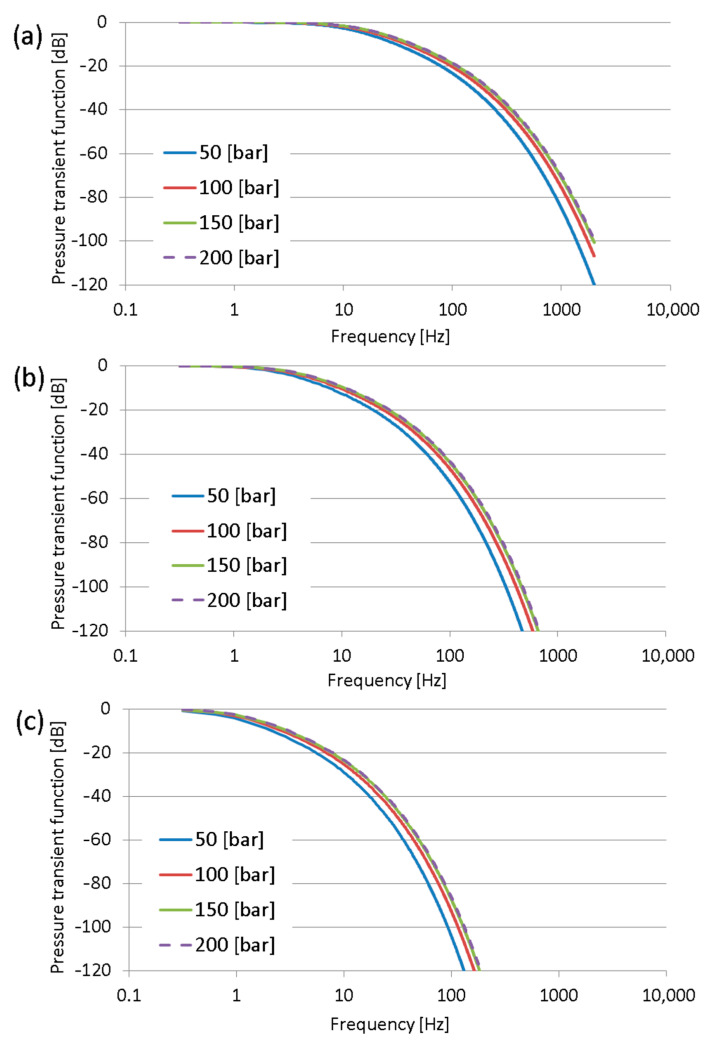
Pressure transfer functions of DN2 measuring hoses of length: (**a**) 0.4 m; (**b**) 0.8 m; and (**c**) 1.5 m.

**Figure 4 sensors-23-07056-f004:**
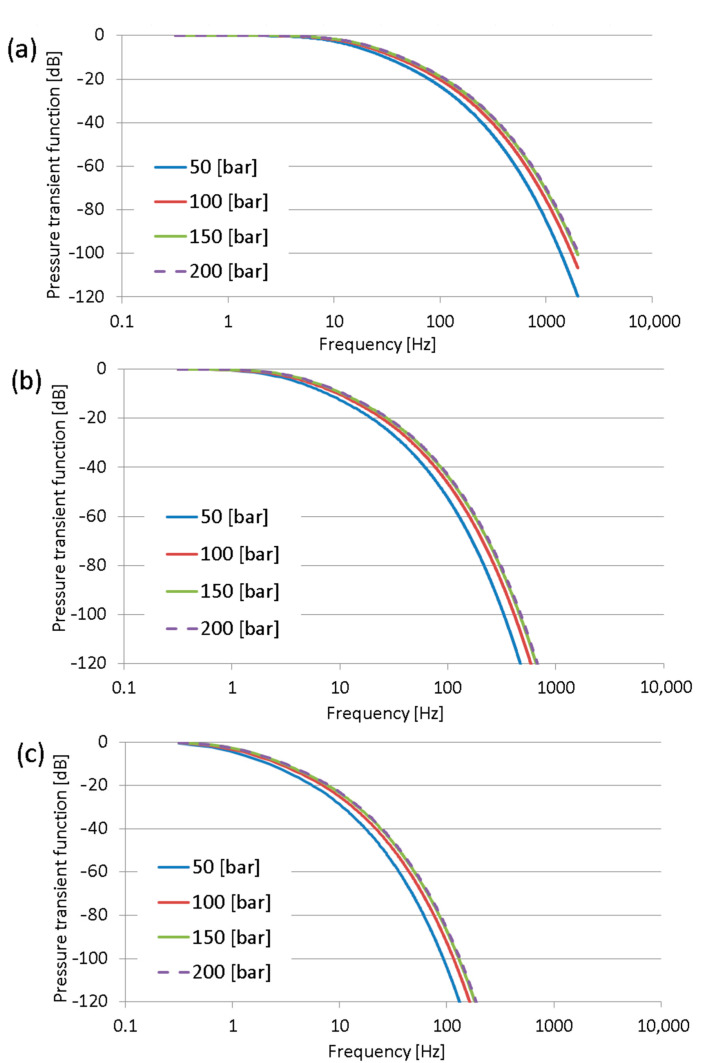
Pressure transfer functions of a DN4 hose with blocked output, length of hose: (**a**) 0.4 m; (**b**) 0.8 m; and (**c**) 1.5 m.

**Figure 5 sensors-23-07056-f005:**
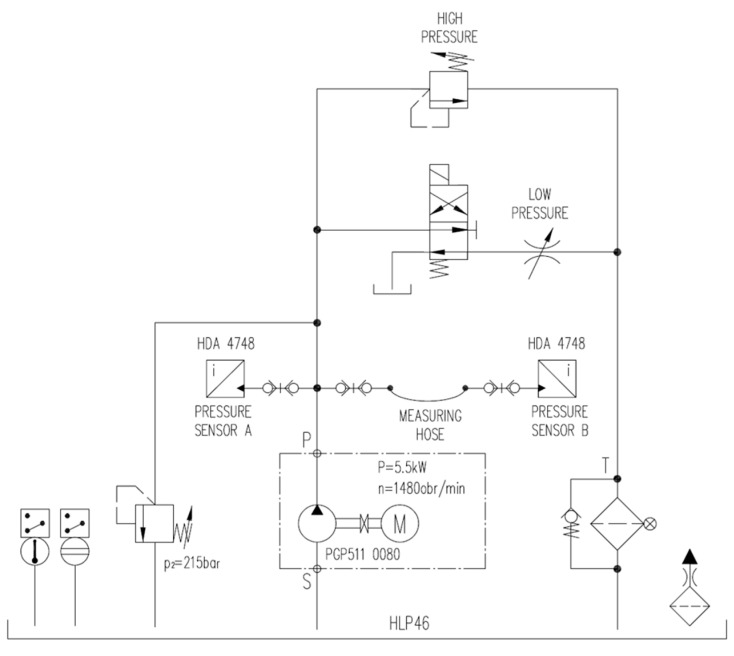
Simplified hydraulic diagram of a test stand for preliminary research.

**Figure 6 sensors-23-07056-f006:**
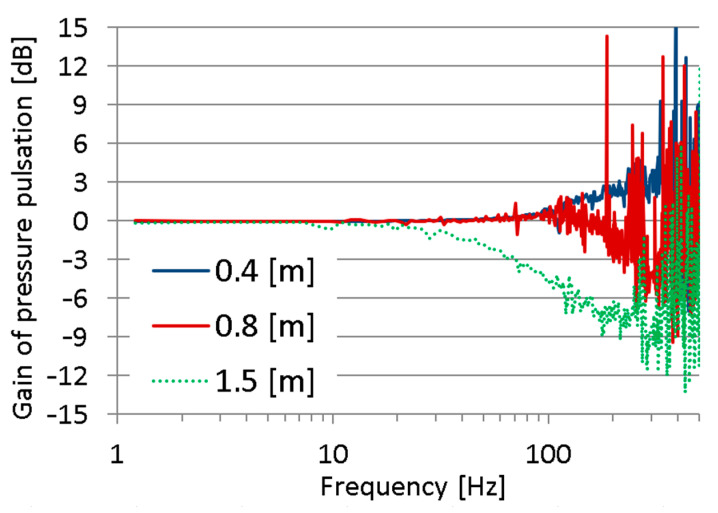
Gain of measuring hoses of 0.4 m, 0.8 m, and 1.5 m length for 100 bar average pressure.

**Figure 7 sensors-23-07056-f007:**
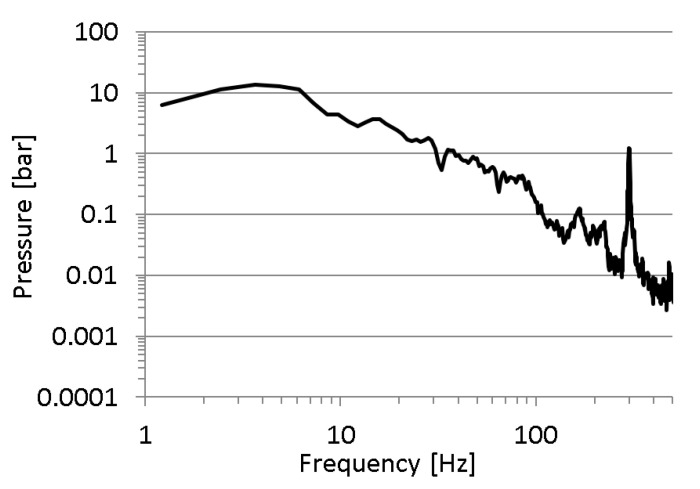
Spectrum of the excitation signal (namely the pressure in a discharge line) for the average pressure of 100 bar.

**Figure 8 sensors-23-07056-f008:**
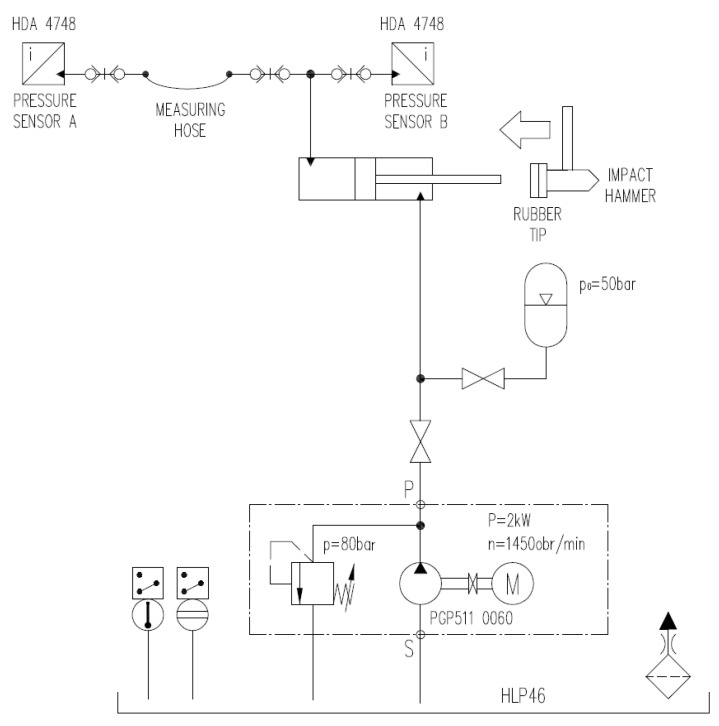
Simplified hydraulic diagram of the test stand for impact tests.

**Figure 9 sensors-23-07056-f009:**
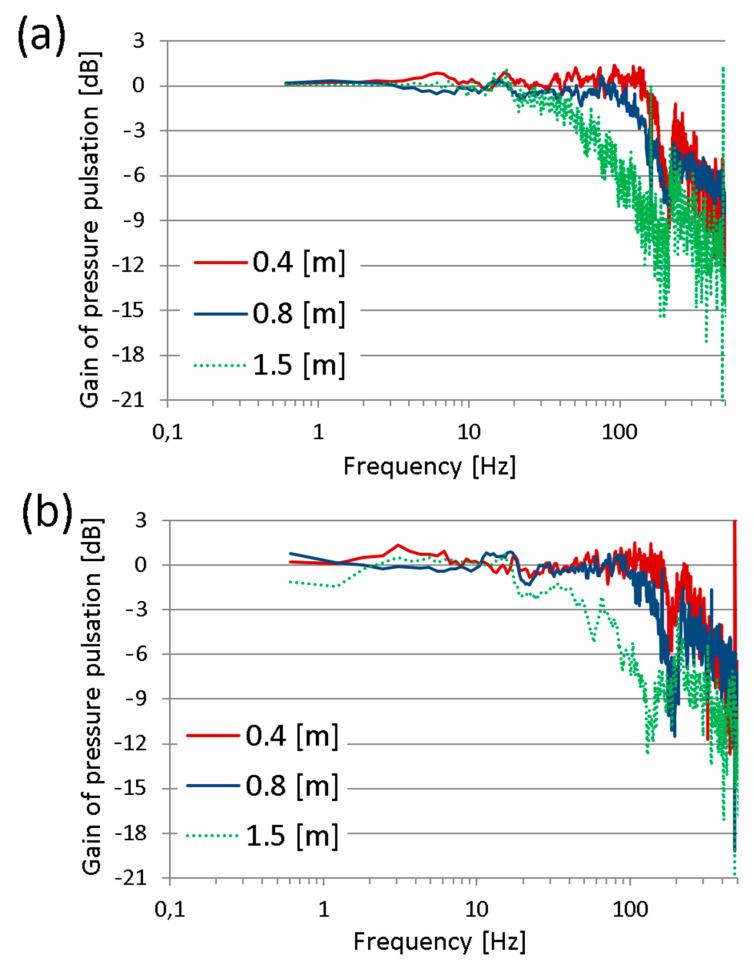
Gain of measuring hoses 0.4 m, 0.8 m, and 1.5 m in length for the hammer test: (**a**) hard tip and (**b**) soft tip.

**Figure 10 sensors-23-07056-f010:**
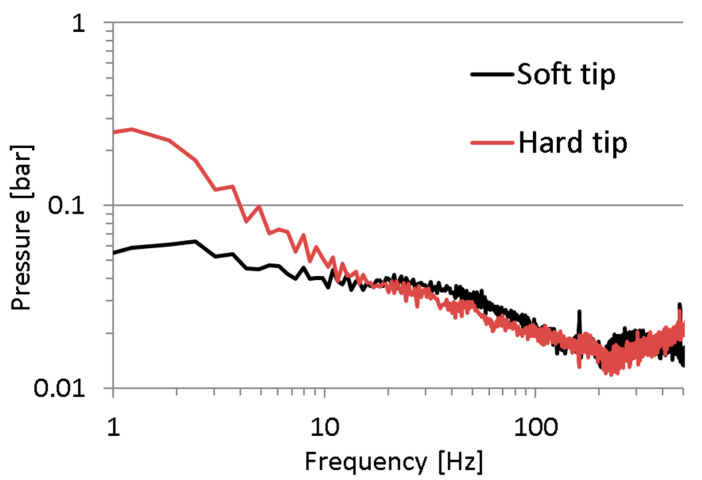
Spectrum of excitation signal during the Impact Hammer Test.

## Data Availability

Detailed data on the conducted studies are available from the corresponding author.

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
