# Peer review of "Hydraulic Measuring Hoses as Pressure Signal Distortion—Mathematical Model and Results of Experimental Tests"

_sensors, 2023, doi:10.3390/s23167056_

Round 1
Reviewer 1 Report
This manuscript investigates a hydraulic signal hose on the change of indications of the pressure transducer in high dynamics of operation hydrostatic drives and controls. The experimental results are provided to illustrate the effectiveness of the proposed method. Hence, the manuscript is interesting and the presentation is well-written.
The author can refer to the following comments, and improve the contribution and presentation in the revision.
(1) How to construct the dynamic model of hydraulic signal hose in distributed condition, since many distributed model should be considered in this study.
(2) Figure 3 shows the pressure transfer functions of DN2 measuring hose of length. How to illustrate the pressure affect to the system?
(3) Some chatters also exist in the gain of measuring hoses of different length as shown in Fig. 6. Please explain the reason in that case.
(4) Some new references also discussed hydraulic control similar to the author’s idea, such as neural adaptive backstepping control of a robotic manipulator, and parametric adaptive control of single-rod electrohydraulic system. The authors may give more descriptions about related work.
(5) There exist many typos and grammar errors in text. Please carefully check all the presentations in the revision.
None.
Reviewer 2 Report
In this manuscript, the influence of Minimess® hydraulic signal hose on the change of pressure sensor indication under highly dynamic working conditions is modeled and experimentally verified. It is noted further that the values of cut-off frequency of the measurement hoses non-linearly depend on their lengths. Generally, the logic of the paper is clear and the work is interesting. So, we recommend that the manuscript can be published after finishing some modifications.
Some suggestions are listed as follows:
1. Please carefully check the spelling of words in the manuscript, some words are misspelled.
2. In the first section, it is mentioned that "This is especially important for users of highly dynamic hydraulic systems consisting of servo valves or proportional control valves", please add the possible influence of the measuring hose on the accuracy of pressure pulsation measurement and why this is important for users of high-dynamic hydraulic systems. This will better reflect the practical engineering application of the authors' work and improve the readability and novelty of the manuscript.
3. In the second part of the manuscript, the pressure transducer is mounted on the output port of the two-port network model, and the volume flow rate “qout” of the output port is ignored, does this have any effect on the dynamic phenomenon of the hydraulic hose?
4. The manuscript mentions that the signal direction is opposite to the fluid flow direction in long hydraulic transmission lines, but why is the signal direction the same as the fluid flow direction in the measuring hose?
5. In the second section of the manuscript, it is mentioned that "The inner radius of a hose is 1 mm, the kinematic viscosity was accepted as well as the viscosity of HLP46 fluid at 30EC". What do the terms "HLP46" and "30EC" stand for? Please provide the full name of the terms. Also, this case is for 1mm inner radius, what is the case when the inner diameter of the measuring hose is 4 mm (2 mm inner radius)?
6. The DN2 and DN4 measuring hose were analyzed in the mathematical model, but only the DN2 measuring hose was selected for the experimental test, why was the DN4 measuring hose not tested in the experiment?
Some words are misspelled and need to be corrected and carefully checked.
Reviewer 3 Report
1. Abstract
|
Line/Eq./Fig. |
Remark |
|
general |
Bad quality of English/incomprehensible language |
|
|
|
|
|
|
|
|
|
|
|
|
|
|
|
|
|
|
2. Introduction
|
Line/Eq./Fig. |
Remark |
|
35 and 40 |
repetitive |
|
34 and 45 |
Far too broad and general citing, repetitve |
|
52 |
Add citation to minimess catalogue |
|
58-61 |
Citations are simply given without relating to them. Give at least a minimal insight in what is referred to in these citations |
|
General |
Topic technically interesting but presumably scientifically not very creative and enligthing |
|
|
|
|
|
|
3. Dynamic phenomena occurring in hydraulic lines
|
Line/Eq./Fig. |
Remark |
|
65 and 98 |
Wrong numbering of chapters |
|
|
To what extent are the contents of this article duplicates of the cite: Kudźma, Z. Damping pressure and noise pulsation in transient and established conditions in hydraulic systems. Publishing 410 House of Wroclaw University,Poland, 2012 (in Polish).? Unfortunately, I do not have access to the publication. |
|
|
|
|
|
|
|
|
|
|
|
|
|
|
|
4. Mathematical model
|
Line/Eq./Fig. |
Remark |
|
Eq. 3 |
Redundant; combine with Eq. 2 or simply state that H is square and of dim 2x2 |
|
160 |
Different greek letter for density compared to line 153 |
|
General |
Subscripts that do not signify running indices are written upright and not italic |
|
Fig. 1 and Fig. 2 |
Replace with higher resolution file or even better with vector graphic |
|
174 |
Misplaced indent at BOL |
|
176 |
I am not sure if I understood this correctly, but even if q_out is zero there still might be an - admittedly small - q_in resulting in pressure build up inside the pipe. Therefore, you would need to consider h21 as well. Is this neglected regarding the small volume of the pipe? |
|
198-200 |
Unclear which diameter is used—1 mm or 2mm? |
|
201 |
EC->°C |
|
General |
Own section for simulation results. Do not mix up mathematical theory with simulations or experiments |
|
Fig. 3 |
Replace with higher resolution file or even better with vector graphic |
|
214 following |
Misplaced indent at BOL |
|
Fig. 4 |
Replace with higher resolution file or even better with vector graphic |
|
219 |
Instead of amplitude fall use “-3 dB” |
|
234-236 |
“The slope for the first range of each curve of the pressure 234 transfer function equals to -20 dB/decade” What is the definition of the “first range”? For me this frequency response looks like the one of a delay (dead time) and not a first order low pass filter. |
|
245-247 |
“The calculation results for models of the measuring hoses based on the mathematical model of a hydraulic long transmission line are very controversial and therefore require an experimental verification” What do you mean by controversial? Do the results contradict the theoretical expectations? What exactly were the expectations? Are the results controversial to general expectations or common knowledge? |
|
|
|
|
|
|
|
|
|
5. Assumptions of experimental test
|
Line/Eq./Fig. |
Remark |
|
248 |
Wrong numbering of chapters |
|
250 - 251 |
The ideal signal for a frequency response is a constant aplitude sweep signal as contrary to a dirac or step the amplitude of the input is constant over all included frequencies whereas amplitudes fall dramatically below noise level for higher frequencies at a dirac or step. See: https://www.systemwissenschaften.de/systemidentifikation-mit-sprungantworten-teil-i-das-problem (I apologize that the site is only in german, but you might use one of the popular translator programs to get the meaning if you do not speak german) |
|
252-258 |
Revision of language necessary. Bad quality of language. Difficult to understand. Unclear which two methods/types are distinguished here. Please clarify. |
|
258 |
“This is a very costly solution.” Too colloquial, please rephrase |
|
|
|
|
|
|
|
|
|
6. Periodic changes in pressure on discharge line as excitation signal
|
Line/Eq./Fig. |
Remark |
|
260 - 261 |
What about rotary slide valves? You could vary the frequency and amplitude as well |
|
277-280; 285;287-291;301 – 305; … |
Misplaced indent at BOL |
|
288 |
What is the decision criteria of the window shape and length? |
|
293 |
Subscripts that do not signify running indices are written upright and not italic |
|
265 - 269 |
Why don’t you use two pressure relief valve for two pressure levels? Is the throttle varied during tests? |
|
302-304 |
Did you evaluate the noise levels of the pressure sensors at steady state? Did you consider quantization errors? |
|
Fig. 7 |
See remark to 250 – 251; Amplitude over 100 Hz too low |
|
|
|
|
|
|
7. Impact hammer test for measuring hoses
|
Line/Eq./Fig. |
Remark |
|
354 |
Wrong reference number for figure |
|
356 |
What is the decision criteria of the window shape and length? |
|
General |
Analyzation of the frequency response of the hammer (for comparison with previous excitation signal of figure 7) is missing |
|
375 -376 |
Could it be that the amplification effect at short pipe lengths results from resonance effects when the pressure oscillations are reflected at the closed end of the pipe. These effects would accumulate during sinusoidal excitation over some time whereas the impulse hammer excites the resonance frequency only in a very short time span. |
|
375-379 |
I don’t really get this reasoning- You stated that the measuring hoses act like a low pass filter, but the assumption that they damp pressure pulsations were false? In my opinion, this reasoning is not consistent and contradictory. |
|
|
|
|
|
|
|
|
|
I would advise to investigate resonance effects in the pipes - especially short pipes - at harmonic excitation and use sweep signals as exctation e.g. by using a rotary slide valve with variable frequency.
Furthermore, the effect of compression flow is neglected in the mathematical model i.e. p_in > 0 @ p_out = 0. You should at least justify this by a small numerical study.
Overall, this topic is of some practical interest to application tasks but the scientific quality and value in this state is too low for publication. Rejection of this paper might be revoked if additional experiments are carried out and the potential omissions in the theoretical background are eliminated.
Please have a look at the notes, there are not only technical and formal notes but also comments on the quality of english.
Reviewer 4 Report
Comments to the Authors
The distortion of hydraulic measuring hose as pressure signal is a very common problem in scientific research experiments, and the related research on this is also of great significance. I have a few suggestions for your article:
1. The diagram in the article is of poor quality, please improve it. For example, the structure of the diagram in Fig 5 and 8 is not clear, and readers cannot intuitively understand the true meaning of the author. Fig 6 and 9, the picture is not clear. The font sizes of 'a and b ' in Fig. 4 and Fig. 9 are different, so the standard should be unified.
2. Article subtitle sorting error, please correct.
3. Please explain what is said in the article: "It follows that the assumption that pressure pulsations were strongly damped by measuring hoses is not true.", can you further analyze the reasons for its error here?
4. It is stated in the conclusion that the cut-off frequency of the DN2 hose is extremely low, and one of the potential reasons for this result may be the inappropriate setting of the mathematical model of the hydraulic pipeline. How do you verify the reliability and feasibility of the mathematical model you set。
5. Conclusion It is emphasized that the cut-off frequency value of the measuring hose depends on its length nonlinearly. Could you please further give the relationship between the cut-off frequency and length of the hose?
6. This paper presents a mathematical model and experimental study of the influence of pressure signal hose on the change of pressure sensor indication in drive and control. This study is relatively simple, the conclusion is also relatively simple. Please explain the mechanism in depth and deepen the core of the article.
The English does not need to be modified
Reviewer 5 Report
The author's of manuscript proposes the basic concept of Minimess® hydraulic signal hose influence on the change of indications of the pressure transducer in high dynamics of operation hydrostatic drives and controls.
The manuscript is clear, relevant for the field, a gap in knowledge is identified - the research studies on evaluation hydraulic measuring hoses as pressure signals distortion not enough. Model and experimental studies show that without any limits the measuring hoses can only be used to measure the average value of the pressure. In the case of measuring pressure waveforms the relationship between the length of the measuring hose and its cut-off frequency should be known. The methods and experimental verification systems are appropriate.
The results and conclusions are actual and the scientific impact of the manuscript is not very high, because here is no comparison with findings of other authors. More then half literature sources in reference list are from Poland, but the problem mentioned in manuscript is actual worldwide. The findings and their implications should be discussed in the broadest possible context.
I would recommend to collect all questions from the manuscript subchapters and resolve them in the chapter "Discussion" before "Conclusions".
Future research directions may also be highlighted.
Round 2
Reviewer 2 Report
The manuscript has been revised as required and can be accepted.
Author Response
The authors thank for positive review.
Reviewer 3 Report
Version 2
1. General
|
Line/Eq./Fig. |
Remark |
|
format |
Layout error in header (line broken) |
|
format |
Use full justification as text layout not “flush left” |
|
format |
Beginning of line (BOL) often not flushed but ragged with misplaced indents |
|
|
|
|
|
|
|
|
|
|
|
|
2. Abstract
|
Line/Eq./Fig. |
Remark |
|
general |
Bad quality of English/incomprehensible language |
|
|
|
|
|
|
|
|
|
|
|
|
|
|
|
|
|
|
3. Introduction
|
Line/Eq./Fig. |
Remark |
|
35 and 40 |
repetitive |
|
52 |
Add citation to minimess catalogue |
|
58-61 |
Citations are simply given without relating to them. Give at least a minimal insight in what is referred to in these citations |
|
|
|
|
|
|
4. Dynamic phenomena occurring in hydraulic lines
|
Line/Eq./Fig. |
Remark |
|
|
To what extent are the contents of this article duplicates of the cite: Kudźma, Z. Damping pressure and noise pulsation in transient and established conditions in hydraulic systems. Publishing 410 House of Wroclaw University,Poland, 2012 (in Polish).? Unfortunately, I do not have access to the publication. |
|
|
|
|
|
|
|
|
|
|
|
|
|
|
|
5. Mathematical model
|
Line/Eq./Fig. |
Remark |
|
80 & 113 |
Same section number |
|
Eq. 3 |
Redundant; combine with Eq. 2 or simply state that H is square and of dim 2x2 |
|
Fig. 1 and Fig. 2 |
Replace with higher resolution file or even better with vector graphic |
|
179 |
Misplaced indent at BOL |
|
188 |
I am not sure if I understood this correctly, but even if q_out is zero there still might be an - admittedly small - q_in resulting in pressure build up inside the pipe. Therefore, you would need to consider h21 as well. Is this neglected regarding the small volume of the pipe? Further clarification necessary! Why can one neglect cross-influences? |
|
General |
Own section for simulation results. Do not mix up mathematical theory with simulations or experiments |
|
Fig. 3 |
Replace with higher resolution file or even better with vector graphic |
|
218 |
Misplaced indent at BOL |
|
Fig. 4 |
Replace with higher resolution file or even better with vector graphic |
|
|
|
|
|
|
|
|
|
6. Assumptions of experimental test
|
Line/Eq./Fig. |
Remark |
|
268 following |
Revision of language necessary. Bad quality of language. Difficult to understand. Unclear which two methods/types are distinguished here. Please clarify “high workload” |
|
|
|
|
|
|
|
|
|
7. Periodic changes in pressure on discharge line as excitation signal
|
Line/Eq./Fig. |
Remark |
|
274 |
“What is the MOST economic …” |
|
310 |
Subscripts that do not signify running indices are written upright and not italic e.g. |
|
|
|
|
|
|
8. Impact hammer test for measuring hoses
|
Line/Eq./Fig. |
Remark |
|
General; 378-380 |
Analyzation of the frequency response of the hammer (for comparison with previous excitation signal of figure 7) is missing |
|
Fig.6 & Fig.7 |
Could it be that the amplification effect at short pipe lengths results from resonance effects when the pressure oscillations are reflected at the closed end of the pipe. These effects would accumulate during sinusoidal excitation over some time whereas the impulse hammer excites the resonance frequency only in a very short time span. |
|
398-400 |
I don’t really get this reasoning- You stated that the measuring hoses act like a low pass filter, but the assumption that they damp pressure pulsations were false? In my opinion, this reasoning is not consistent and contradictory. |
|
|
|
|
|
|
|
|
|
I would advise to investigate resonance effects in the pipes - especially short pipes - at harmonic excitation and use sweep signals as excitation e.g. by using a rotary slide valve with variable frequency.
Furthermore, the effect of compression flow is neglected in the mathematical model, i.e. pin > 0 @ pout = 0. You should at least justify this by a small numerical study.
In the second version of the article the authors achieved a better quality of language and readability. Also, some – but not all – formatting errors were corrected. The rebuttal letter answered most of my questions, but actual changes to the studies procedure, i.e. experiments with different components, further simulative studies with more sophisticated models, analysis of resonance effects at harmonic excitation, were not performed. I understand that additional experiments and new components are costly and time-consuming contradicting the available time window in such a review-process, but I strongly advise to perform them or at least consider them in a follow up study to underpin the results and increase the overall quality.
Score: Accept with major revision necessary.
Quality of english improved in version 2 and article is far better readible now.
